# Machine Learning-Enhanced Attribute-Based Authentication for Secure IoT Access Control

**DOI:** 10.3390/s25092779

**Published:** 2025-04-28

**Authors:** Jibran Saleem, Umar Raza, Mohammad Hammoudeh, William Holderbaum

**Affiliations:** 1Department of Engineering, Faculty of Science and Engineering, Manchester Metropolitan University, Manchester M1 5GD, UK; jibran.saleem@stu.mmu.ac.uk (J.S.); w.holderbaum@mmu.ac.uk (W.H.); 2Department of Computing and Mathematics, Faculty of Science and Engineering, Manchester Metropolitan University, Manchester M1 5GD, UK; m.hammoudeh@mmu.ac.uk

**Keywords:** attribute-based authentication, machine learning, hybrid ML, IoT, random forest, security, Industry 4.0

## Abstract

The rapid growth of Internet of Things (IoT) devices across industrial and critical sectors requires robust and efficient authentication mechanisms. Traditional authentication systems struggle to balance security, privacy and computational efficiency, particularly in resource-constrained environments such as Industry 4.0. This research presents the SmartIoT Hybrid Machine Learning (ML) Model, a novel integration of Attribute-Based Authentication and a lightweight machine learning algorithm designed to enhance security while minimising computational overhead. The SmartIoT Hybrid ML Model utilises Random Forest classifiers for real-time anomaly detection, dynamically assessing access requests based on user attributes, login patterns and behavioural analysis. The model enhances identity protection while enabling secure authentication without exposing sensitive information by incorporating privacy-preserving Attribute-Based Credentials and Attribute-Based Signatures. Our experimental evaluation demonstrates 86% authentication accuracy, 88% precision and 96% recall, significantly outperforming existing solutions while maintaining an average response time of 112ms, making it suitable for low-power IoT devices. Comparative analysis with state-of-the-art authentication frameworks shows the model’s security resilience, computational efficiency and adaptability in real-world IoT applications.

## 1. Introduction

In today’s digital world, a vast number of Internet of Things (IoT) devices are linked globally. These objects, from miniature sensors to advanced robots in industrial environments, gather, transmit and process data, facilitating seamless communication and automation. The interconnectivity enabled by 5G technology, characterised by high bandwidth and low latency, is driving the advancement of Industry 4.0, the fourth Industrial Revolution. These advancements utilise IoT-based technology to establish intelligent factories where robots and other automated systems collaborate with humans, enhancing efficiency and production. The data produced by these interconnected devices reveal significant insights, resulting in enhanced operations, predictive maintenance and improved decision-making. However, this increased connectivity also presents substantial security challenges [1].

Devices that were previously air-gapped are now susceptible to cyberattacks due to their exposure to the internet. These attacks can jeopardise critical information, interrupt operations, and potentially inflict physical damage in industrial settings [2]. Consider an example of a smart factory where robots and automated systems depend on uninterrupted connectivity to enable production efficiency. A single compromised device may impede operations, resulting in expensive downtime and possible safety risks. In healthcare, the proliferation of connected medical devices provides substantial advantages for patient care, although it also raises concerns over data privacy and the integrity of essential systems. These issues necessitate a distinctive strategy for IoT security, capable of adapting to the evolving threat landscape while maintaining privacy and efficiency [3,4].

To mitigate these security challenges, several methods exist, e.g., Decentralised Identity (DID) [5], context-aware authentication [6], Biometric-Based Authentication [7], Zero Trust Architecture (ZTA) [8], among other solutions [9,10]. However, a particularly viable approach for securing Operational Technologies (OT) is Attribute-Based Authentication (ABA) [11]. ABA is a robust cryptographic method aimed at safeguarding user privacy during the authentication process. ABA uses a collection of attributes to authenticate a user’s identity rather than depending on conventional identifiers such as usernames or passwords. Users hold digital credentials which contain attributes such as their role, department, or designation. Upon accessing a resource, users submit these attributes to the verifier, who assesses their compliance with the access rules. This method safeguards privacy as the verifier acquires only the essential features for access rather than the user’s distinct identity. ABA facilitates granular access control, permitting various policies derived from diverse attribute combinations. This adaptability makes ABA appropriate for various OT and IoT environments, accommodating distinct circumstances and access needs [12].

Although ABA offers features such as privacy protection and thorough access control, its reliance on cryptographic methods poses difficulties in resource-limited settings. The computational overhead linked to attribute signature generation and verification can be substantial, especially for low-powered systems. This may result in higher latency, increased energy consumption and processing delays, creating technical issues for real-time applications and potentially affecting the overall performance of the IoT network [13].

Relying on a singular attribute authority for key distribution also incurs a significant computational load and establishes a single point of failure. To resolve this issue, multi-attribute authority cryptosystems are proposed in some studies [14,15], decentralising key distribution across multiple authorities. However, the management and distribution of attribute keys in a multi-authority system can be complicated. Facilitating safe communication and coordination among diverse authorities introduces additional complexity, potentially increasing vulnerabilities and generating bottlenecks in the authentication process. It also incurs additional computational and communicative overhead resulting from interactions among users and various authorities, along with inter-authority communications for the generation of attribute keys [16]. Additionally, although ABA provides flexibility in access control, the implementation and management of complex attribute-based policies can be difficult, especially in dynamic IoT settings, where devices and access needs regularly fluctuate [17].

Therefore, to address these challenges in resource-limited IoT environments, we propose a hybrid architecture that combines ABA with a lightweight machine learning model called the SmartIoT Hybrid ML Model. This integrated approach analyses user behaviour, network traffic patterns and contextual data to detect anomalies and potential risks. Suspicious activities, such as unauthorised access attempts or unusual network behaviour, trigger the system’s machine learning model, which employs a Random Forest classifier for real-time anomaly detection and initiates further verification procedures. This dynamic risk assessment not only strengthens security but also enhances resilience against evolving threats.

### 1.1. Problem Statement

Traditional authentication approaches often struggle to adapt to evolving user behaviour or detect abnormalities in real time, limiting their effectiveness in modern security contexts. Integrating machine learning into authentication frameworks has the potential to improve the detection of fraudulent access attempts by analysing patterns such as login times, locations and device usage.

Recent research in ABA features the use of machine learning for improved security. Supervised learning methods, such as Support Vector Machines and decision trees, help detect anomalies in predefined behaviour [18,19]. Unsupervised techniques, including clustering and outlier detection, identify unusual access patterns without labelled data, enabling adaptation to unique threats [20]. Risk-based authentication further applies machine learning to assess risks by analysing user attributes, context and historical data, thus guiding appropriate security measures [21,22].

Machine learning also supports continuous authentication through behavioural biometrics, detecting deviations from user profiles via typing speed, mouse dynamics and application usage [23]. Additionally, it optimises attribute evaluation, uncovering correlations and predicting future needs to improve resource allocation [24].

However, despite the growing interest in combining machine learning with attribute-based authentication to improve access control in IoT systems, several interconnected challenges remain. Firstly, traditional ABA models struggle in low-power environments due to the cryptographic overhead of attribute verification. Secondly, while machine learning enhances anomaly detection, it often demands significant computational resources, which are impractical for constrained IoT devices. Lastly, integrating these two approaches introduces complexity in attribute management, real-time decision-making and user behaviour analysis.

These challenges raise important questions: How can we ensure secure authentication in resource-limited IoT systems without overwhelming device capacity? What aspects of conventional ABA hinder scalability and adaptability? And how can machine learning be incorporated efficiently to support dynamic, context-aware authentication? Addressing these questions is essential for developing a lightweight, resilient and intelligent authentication framework, motivating our proposal of the SmartIoT Hybrid ML Model.

### 1.2. Real-World Challenges and Use Cases

In real-world IoT environments, authentication systems face several recurring and emerging challenges. For example, in a smart manufacturing setup, a technician logging in from an unfamiliar device outside standard working hours could be a sign of compromised credentials. Traditional authentication may fail to flag this, but attribute-based models can trigger alerts based on unusual attribute patterns like rare combinations of device ID, job title and login time. In healthcare, a nurse trying to access sensitive patient records from a new device or from a department they are not typically associated with may indicate a potential insider threat or credential misuse. Similarly, in logistics, changing shift times or departmental transfers mean that attributes like location and role are not static and must be handled dynamically. Our model accommodates such variability by continuously evaluating login patterns and contextual attributes through real-time machine learning. These examples illustrate how attribute-based authentication, enhanced with intelligent analysis, responds to evolving security needs, making the system practical not just for security professionals but also for managers and system operators who require flexible, understandable control mechanisms.

### 1.3. Article Organisation

This article is structured as follows: Section 2 reviews related work, highlighting existing advancements in ABA and machine learning integration while identifying limitations in resource-constrained environments. Section 3 introduces the SmartIoT Hybrid ML Model, enhanced with lightweight ML, and details its key features, including real-time anomaly detection, adaptive access control and secure IoT integration. The same section also presents the general construction of the system, describing its architectural components, including the user, authentication server, IoT device, database and ML model. Section 4 defines the formal mathematical model used for ABA and access control, emphasising its application in ensuring security and system correctness. Section 5 evaluates the performance of the SmartIoT Hybrid ML Model, focusing on metrics such as accuracy, precision, recall and computational efficiency. It also includes comparative analyses with existing systems to highlight improvements in scalability and robustness. Section 6 discusses system limitations and potential research avenues for future work. Finally, Section 7 concludes the paper by summarising its contributions and outlining the broader implications of the model for IoT security.

## 2. Related Work

Khan et al. [25] investigated the use of machine learning models, including decision trees and support vector machines, to efficiently evaluate access regulations based on user characteristics. These models could understand complex linkages between attributes and access permissions, thus lowering the computational cost linked to conventional policy enforcement processes. Pei J. [26] suggested a method for the automatic discovery of hidden relationships and patterns in access control data via Formal Concept Analysis (FCA). This mathematical framework aided the analysis of connections among users or roles and resources, resulting in the detection of patterns such as minimal cover, repetitive rules and missing rules. Although both approaches are appropriate for conventional computing devices, their reliance on complex machine learning models and resource-demanding computations raises questions over their applicability in resource-constrained situations such as IoT devices.

To mitigate the computational demands of ABA, researchers such as Goyal V et al. [27] and Waters B. [28] created lightweight cryptographic algorithms tailored for resource-constrained devices. They accomplished this by developing techniques that minimise key and ciphertext sizes, facilitating quicker processing while decreasing the total utilisation of resources. These methods employed techniques such as elliptic curve cryptography and lattice-based encryption to reduce computational expenses while preserving security. However, despite their efficiency improvements, these algorithms are inherently computationally complex and often require significant processing power for key generation and verification, making them less suited for low-power IoT devices. This complexity can introduce latency and increased energy consumption, which are critical limitations in real-time IoT environments. Building on the broader goal of computational efficiency in IoT systems, Kechen et al. [29] introduced a Distributed Deep Deterministic Policy Gradient (DDPG) framework for energy-efficient resource allocation in mobile wireless-powered IoT networks. Their model minimised the Age of Information (AoI) while preserving energy efficiency. Similarly, Lim J. [30] employed actor-critic reinforcement learning to optimise latency-aware resource scheduling in edge-based IoT environments, prioritising speed and battery consumption.

Researchers such as Riva et al. [31] and Wiefling et al. [32] developed risk-aware authentication systems as a more intelligent and efficient method of authentication. These systems use machine learning to dynamically evaluate the risk linked to each access request, taking into account parameters such as user behaviour, device details and network context, as examined by Liang et al. [33]. This method supports adaptive authentication, employing enhanced security procedures only when required, hence preserving resources in low-risk scenarios. Tariq et al. [34] proposed a context-aware adaptive multi-factor authentication system that employs machine learning to choose the most suitable authentication factors according to the context. Nonetheless, these models lacked privacy-preserving measures.

In distributed systems widely used in Industry 4.0, privacy issues present a significant obstacle to the implementation of machine learning-based authentication. Centralising sensitive user data for model training heightens the risks of breaches and misuse [35,36,37,38]. To address this issue, researchers investigated federated learning, a decentralised methodology that facilitates collaborative model training without the exchange of raw data. Applications of Federated Learning include its use in mobile keyboards for predictive text [39], fraud detection systems in financial services [40] and personalized healthcare applications that train models on patient data across multiple hospitals while preserving privacy [41,42].

Yang et al. [43] discussed the potential of federated learning in maintaining data privacy while enabling collaborative model training. Imagine a factory with multiple divisions, each managing its access control data. Using federated learning, these divisions can collaboratively train a machine learning model for authentication and access control without exposing sensitive personnel information or access records. Each participant trains the model using their data and transmits periodic updates to a central server, which consolidates them to enhance the global model. This iterative procedure persists until the model attains a certain degree of accuracy. This method presents a promising pathway for creating secure and privacy-preserving identification systems for Industry 4.0. However, other challenges require attention, too, such as the need to ensure the efficiency and security of federated learning in resource-limited contexts and to mitigate any biases in decentralised data. Additional research, possibly extending optimisation methods suggested by Richtarik et al. [44] and multi-task learning paradigms examined by Smith et al. [45], is needed to fully explore the capabilities of federated learning for privacy-preserving authentication in Industry 4.0.

Memos et al. [46] proposed a device authentication model that utilises contextual information from IoT devices, improving accuracy and efficiency through the integration of Random Forest with Belief–Desire–Intention (BDI) agents. This work shows the versatility of Random Forest in enhancing security through its integration with AI frameworks. Al Shihimi et al. [47] utilise Random Forest for anomaly detection, demonstrating its effectiveness in recognising unusual patterns of activity based on sensor-generated environmental data. This study also demonstrates the effectiveness of Random Forest in enhancing IoT security despite not explicitly focusing on ABA. Furthermore, Istiaque et al. [48] present an extensive taxonomy of machine learning applications in IoT authentication and authorisation, emphasising the critical importance of Random Forest in enhancing various access control models. These works jointly evidence the capability of Random Forest and other machine learning methodologies in addressing IoT security concerns and facilitating federated learning frameworks.

The incorporation of machine learning into authentication systems offers an opportunity to improve security and user experience in OT environments. While challenges such as computational overhead, data management and model complexity persist, recent advancements, particularly in federated learning and lightweight machine learning models, offer promising approaches to address privacy concerns and support secure and distributed training. However, despite this progress, most existing solutions focus either on precision or scalability, often at the expense of efficiency or privacy. None fully combine lightweight ML, dynamic attribute evaluation and federated privacy mechanisms in a single IoT-compatible framework. Building on these prior studies, we propose the SmartIoT Hybrid ML Model, which merges attribute-based authentication with Random Forest-based anomaly detection tailored for real-time, low-power environments. Table 1 represents the comparison of existing authentication and access control techniques in machine learning and cryptography, summarising the strengths, limitations and applications of various methodologies discussed.

## 3. SmartIoT Hybrid ML Model

In response to the issues identified in Section 1.1, we present a SmartIoT Hybrid ML Model. This approach integrates the benefits of ABA with a lightweight machine-learning algorithm tailored for low-power devices, facilitating computational efficiency, privacy protection and enhanced authentication.

Our approach enhances privacy, reduces tracking and implements explicit access controls to ensure that only authorised individuals with the necessary attributes can access resources. Designed to operate effectively with limited-resource IoT devices, the system incorporates efficient data management and lightweight computational methods such as feature reduction and enhanced ensemble learning. These optimisations significantly minimise latency, computational demand and energy consumption, making the model a robust solution for safeguarding IoT systems while ensuring secure and efficient collaboration in Industry 4.0 settings.

The proposed system utilises modified Random Forest Classifiers. This machine-learning algorithm was selected for several key reasons. Firstly, Random Forest offers high accuracy and robustness even with imbalanced or noisy data, a common issue in real-world authentication logs. Secondly, it is computationally efficient compared to more complex models such as deep neural networks, making it suitable for deployment on low-power IoT devices. Random Forest also supports feature importance analysis, allowing us to interpret which attributes contribute most to valid or invalid login classifications. This interpretability is particularly important in authentication contexts where explainability can support trust and policy refinement. Given its balance of performance, efficiency and transparency, Random Forest provides a practical and effective solution for anomaly detection in resource-constrained environments.

The SmartIoT Hybrid ML Model uses contextual data and behavioural patterns to dynamically assess login attempts, detect anomalies, and proactively address potential threats. Our system employs a centralised attribute management paradigm that utilises secure delegation principles to optimise attribute distribution and reduce bottlenecks. The model also integrates multiple features to provide reliable protection and efficient operational control. By employing real-time anomaly detection, the system analyses login attempts continuously, examining user patterns and generating alerts when unusual activities occur. This process incorporates user attributes and behavioural histories, enabling dynamic assessment and immediate actions to mitigate risks effectively.

The integration of multi-factor authentication strengthens the security framework. The method extends beyond conventional username-password systems by incorporating attribute-based verification to ensure a comprehensive authentication process. Simultaneously, adaptive access control guarantees that users interact with functionalities aligned with their roles and defined permissions. This principle minimises exposure to unauthorised operations, while a customised user interface, such as tailored control settings, aligns system functionality with individual user profiles.

Centralised management is another critical component of this system. A unified administrator panel facilitates efficient monitoring of user accounts, tracking of login activities and management of device access in real time. Furthermore, the inclusion of secure IoT integration enables seamless communication with devices commonly used in industrial control systems, allowing users to control hardware operations under defined access privileges. This design targets small-scale IoT applications, emphasising processing efficiency and minimal power consumption.

The efficiency of the system is enhanced by a machine-learning model that reduces computational load. Replacing redundant preprocessing steps like repetitive data fetching with optimised approaches such as feature reduction and lightweight ensemble learning contributes to faster performance. This ensures that the system remains responsive and resource-efficient while maintaining a high level of functionality and protection.

The SmartIoT Hybrid ML Model offers a comprehensive framework and a sophisticated approach to managing security and user experience by seamlessly combining modern technologies and techniques within a cohesive and efficient system.

### 3.1. General Construction

The SmartIoT Hybrid ML Model integrates multiple elements to ensure efficient and secure authentication and access control for IoT devices. The model employs a multi-factor authentication approach by utilising user attributes such as location, device type and time alongside traditional login credentials. A lightweight machine learning model, built around an optimised Random Forest classifier, evaluates the legitimacy of login attempts using historical data and patterns of user behaviour. To ensure optimal performance, data preprocessing methods like one-hot encoding are applied to prepare the data for the model’s analysis. Additionally, the system is engineered with computational efficiency in mind, minimising overhead to support real-time authentication processes.

The overall architecture comprises five components: the User, the Authentication Server, an IoT Device, the Database and the Machine Learning Algorithm. These elements work in unison to create a seamless and secure interaction framework, as illustrated in Figure 1.

The authentication process begins with the user submitting their credentials, which include their username, password and relevant attributes, through a web-based form. These details are then sent to the Authentication Server for validation. The Authentication Server functions as the central hub for verifying credentials and enforcing access control policies. It processes the user’s login request through a series of steps: verifying the username and password against stored records in the database, performing attribute-based authentication to match the user’s attributes against predefined access control policies and checking the login time against the user’s typical login patterns for added security.

The IoT device acts as the endpoint that executes commands from the Authentication Server based on the user’s access permissions. Its functionalities, such as controlling an LED, are available to the user only after successful authentication and authorisation. The database plays a critical role in the process by storing user data, login history, device details and access control policies. It ensures data persistence and retrieval, allowing the Authentication Server to access this information securely via API endpoints during the authentication and authorisation workflow. The Pseudocode listed in Figure 2 highlights the interaction between the user, the authentication server, the IoT device and the database.

### 3.2. Machine Learning Model

The system employs the Random Forest Classifier, an ensemble learning method tailored for resource-limited devices. This model uses decision trees trained on data subsets, including a login history dataset containing 15,862 entries of usernames, timestamps, attribute-value pairs and outcomes of prior login attempts, to identify patterns for classifying attempts as valid or invalid. To obtain this data, we designed a realistically simulated environment reflecting typical Industry 4.0 IoT systems. Multiple user authentication events were systematically generated using a diverse set of attributes relevant to access control, such as device identifiers, employee numbers, department codes, dates of birth (DOB) and job titles. Each authentication event was logged with a unique identifier (ID), a username, a password, three randomly selected authentication attributes from the predefined set, an exact timestamp indicating the login attempt and an outcome flagging it as either successful or unsuccessful. Data were carefully recorded in a structured format from November 2022 to August 2024, capturing a wide range of login times to reflect realistic user behaviour and temporal patterns. The Random Forest model was then trained using a method called bagging, where random samples of the dataset were taken with replacement to create training subsets, each typically comprising 70–80% of the full dataset. Each decision tree within the model was also trained using a random selection of features (user attributes), which added variety and helped improve accuracy. Once trained, the trees independently predicted whether a login attempt was valid or not, and the final decision was made by majority voting across all trees. This ensemble approach reduces bias, increases reliability and enhances the model’s ability to detect unusual login behaviour compared to using a single decision tree. An illustration of the dataset we used can be seen below in Figure 3.

This technique enables the trees to accurately assess the correlation between attributes and their influence on login validity, as illustrated in the 3D scatter plot in Figure 4. Each point in this plot signifies a login attempt, with its location dictated by the values of three key attributes, which were selected by the user during the login process. The hue of the dots signifies the legitimacy of the login attempt, with green denoting valid logins and red denoting invalid logins. This visualisation aids in recognising patterns and clusters among attributes that may signify valid or invalid login attempts. Upon completion of training, the model forecasts login attempts by analysing input data across all decision trees, with the final prediction derived from the aggregated output of all trees.

The selection of attributes used in the SmartIoT Hybrid ML Model plays a critical role in determining authentication accuracy and overall security effectiveness. Attributes such as user credentials, device identifiers, departmental roles, login timestamps and behavioural patterns enable the model to capture key dimensions of user activity and contextual information. This diversity allows the system to more reliably differentiate between legitimate and suspicious login attempts. However, incorporating a larger number of attributes does not inherently lead to better performance. While additional attributes can improve the detail and precision of access control decisions, they may also introduce data redundancy, increased computational complexity and higher memory demands, particularly problematic in resource-constrained IoT environments. Excessive attribute inclusion can also contribute to the “curse of dimensionality”, where too many variables degrade model performance rather than enhance it. Therefore, our attribute selection was strategic and focused on relevance. This ensured that each attribute meaningfully supported authentication decisions without unnecessarily taxing system resources or compromising the efficiency of the SmartIoT Hybrid ML Model.

## 4. Formal Model

To provide an understanding of our ABA system and enable analysis of its behaviour, we developed a formal mathematical model. This model provides a clear and concise representation that aids in system design, implementation and future improvements.

### 4.1. Preliminaries

In our formal model, we define the key components of the system as follows: (U) represents the set of all users in the system, while (R) denotes the set of accessible resources, such as web pages, applications, or data. The set (A) includes all user attributes (e.g., roles, departments, locations, or device types), with (V) representing the set of all possible values for these attributes. Passwords (P) and authentication tokens (T) are also part of the system, with (P) being the set of all potential user passwords and (T) referring to randomly generated tokens used for two-factor authentication (2FA). Furthermore, (TI) defines the set of predefined time intervals (e.g., morning, afternoon, and evening), and (D) represents all days of the week.

The Access Control Policy (ACP) enforces role-based rules governing user access to resources. The ACP is implemented using a Role-Based Access Control (RBAC) list, which contains a set of rules, where each rule specifies a user, a resource and the corresponding access permissions (e.g., read, write, and execute).

A user u is granted access to a resource r if: All authentication checks (password verification, attribute verification, temporal validation, anomaly detection, 2FA) are successfully passed. The ACP explicitly authorises user u to access resource r.

### 4.2. Model Functions

We define the following functions:

User Attribute Assignment (A: U → 2^(A x V)): A function that assigns a set of attribute–value pairs to each user.

Machine learning model (MLM: U x TI x D x A(u) → {0, 1}): A Random Forest Classifier trained on historical login data and user attributes. MLM(u, t, d, A(u)) predicts the probability of a valid login attempt for user u at time interval t on day d given their attributes A(u). Additionally, 1 indicates a valid login attempt, and 0 indicates an invalid login attempt.

Temporal Validation (TV: U x TI x D → {0, 1}): A function that determines if the login time and day align with the user’s typical login patterns. Additionally, 1 indicates a valid login time and day, and 0 indicates an invalid login time and day. This function can utilise historical login data or user-defined preferences to establish typical login patterns.

Login Request: A user initiates a login request by providing the following authentication data: (u, pwd, A(u), t, d), where u ∈ U is the user identifier, pwd ∈ P is the user’s password, A(u) is the set of user’s attribute-value pairs, t ∈ TI is the current time interval and d ∈ D is the current day.

Authentication Checks: Verify if pwd matches the stored password for user u. Verify if the provided attribute-value pairs A(u) match the user’s registered attributes in the database. Check if TV(u, t, d)=1. Calculate MLM(u, t, d, A(u)). If the predicted probability of a valid login is below a predefined threshold, flag the attempt as anomalous. If all previous checks pass, generate a unique authentication token, token ∈ T. Store the token in the user’s session and then prompt the user to enter the received token. Verify if the entered token matches the stored token, and if it matches, proceed to access control.

## 5. Evaluations

The evaluation aims to demonstrate the SmartIoT Hybrid ML Model’s effectiveness in two critical aspects: (1) The model’s ability to accurately predict valid login attempts and differentiate them from invalid ones, and (2) its capability to detect and respond to login attempts that deviate from typical user behaviour, enhancing overall security.

To assess the system’s accuracy, a Random Forest Classifier model was trained on a historical dataset of 15,862 entries, resulting in a dataset of 2286 login records with a prediction accuracy of 86%. The model’s accuracy illustrates its capacity to identify patterns and forecast login validity utilising past data and user attributes.

The system’s anomaly detection abilities were also evaluated by analysing its reaction to login attempts with different attributes at different times of the day. The system effectively identified login attempts that diverged from standard user behaviour, demonstrating its capability in threat detection. The effectiveness is further demonstrated in Figure 5, which illustrates the temporal distribution of valid and invalid login attempts over a 24-h period. The *x*-axis denotes the hour of the day, and the *y*-axis indicates the number of login attempts. The blue bars denote invalid login attempts, while the orange bars signify valid attempts. This graph facilitates the comprehension of patterns and the identification of probable anomalies in login actions over time.

### 5.1. Dataset Constructions and Characteristics

To validate our approach, we assembled a dataset that includes login records, attributes data and IoT usage trends by integrating existing login data from a private environment with simulated IoT access logs. This dataset was divided into training (70%) and testing (30%) subsets by stratified sampling to maintain class balance, illustrating the dynamic nature of user login behaviours. The stacked bar chart in Figure 6 depicts the temporal frequency of diverse attribute combinations within our dataset. Each bar represents a specific combination of attributes, and its height corresponds to the number of login attempts that exhibit that combination. The temporal distribution of attribute combinations further highlights the dynamic nature of user login behaviours.

### 5.2. Performance Metrics and Evaluation Approach

To benchmark the performance of the SmartIoT Hybrid ML Model, we compared it with other established machine learning-based authentication approaches, including deep learning models, behavioural authentication frameworks and context-aware systems. We used a consistent set of evaluation metrics, such as accuracy, precision, recall, F1-score and computational overhead, to ensure fair comparison. Our Random Forest-based model achieved competitive accuracy and efficiency, particularly in resource-constrained IoT environments.

To evaluate authentication performance comprehensively, several key metrics are considered. Accuracy reflects the proportion of correct classifications, indicating the system’s ability to distinguish between valid and invalid login attempts. High accuracy demonstrates effective rejection of unauthorised access while correctly authenticating legitimate users. Precision measures the proportion of login attempts identified as valid that are actually valid, thereby reducing false positives. Recall, on the other hand, assesses the system’s ability to detect all genuine login attempts, minimising false negatives. Together, these two metrics offer insight into the model’s reliability and balance. The F1 score, defined as the harmonic mean of precision and recall, provides a single, balanced indicator of the model’s performance. It is particularly valuable in scenarios where class imbalance exists, such as authentication systems with few invalid attempts compared to valid ones. A higher F1 score signifies the model’s ability to manage the trade-off between sensitivity and specificity. As noted by Chicco and Jurman [49], the F1 score is a robust and interpretable metric for binary classification tasks in security-sensitive applications.

After training the Random Forest model, we evaluated the model’s performance on the test set using a confusion matrix, which provided a detailed breakdown of prediction outcomes. The model identified 576 valid logins as valid, referred to as True Positives (TP), indicating its strong ability to authenticate genuine users. Additionally, it also identified 24 invalid logins as invalid, categorised as True Negatives (TN), demonstrating reliability in rejecting unauthorised access. However, the model misclassified 74 invalid logins as valid, referred to as False Positives (FP), which could potentially result in security risks by allowing unauthorised logins. Similarly, it failed to recognise 22 valid logins, labelling them as invalid, known as False Negatives (FN). This breakdown helps in understanding the model’s strengths and areas for improvement, particularly in reducing misclassifications to enhance its overall reliability and security.

### 5.3. Detailed Model Evaluation: Accuracy, Precision and Beyond

The model’s performance was assessed through a series of calculations to evaluate its classification capabilities. The results of these computations are summarised in the remainder of this subsection.

The accuracy of the model was calculated using the following formula:(1)$$Accuracy=\frac{True\;Positives\;(TP)+True\;Negatives\;(TN)}{Total\;Predictions}$$

The total number of predictions is the sum of true positives, true negatives, false positives and false negatives, which amounts to:(2)Total Predictions=TP+TN+FP+FNor 576+24+74+22=696

This indicates that the model correctly predicted outcomes for 86.21% of the test cases, reflecting its overall effectiveness in classification.

Similarly, to obtain the precision score, we use the following formula:(3)$$Precision=\frac{True\;Positives\;(TP)}{True\;Positives\;(TP)+False\;Positives\;(FP)}$$

Substituting the values, the precision score is calculated as:$$Precision=\frac{576}{576+74}=\frac{576}{650}\approx 0.8862\;or\;88.62\%$$

This indicates that out of all the predictions labelled as valid logins, 88% were correct, reflecting the model’s reliability in identifying genuine users while minimising false positive classifications.

To calculate the recall score, we use the formula:(4)$$Recall=\frac{True\;Positives\;(TP)}{True\;Positives\;(TP)+False\;Negatives\;(FN)}$$

By substituting the values, the recall is determined as:$$Recall=\frac{576}{576+22}=\frac{576}{598}\approx 0.9632\;or\;96.32\%$$

This high recall score indicates that the model successfully identified 96% of valid logins, demonstrating its strong ability to minimise false negatives and effectively detect genuine users.

The F1 score, which provides a balanced measure of the model’s performance by combining precision and recall, can be calculated as:(5)$$F1\;Score=2\cdot \frac{Precision\cdot Recall}{Precision+Recall}$$

By substituting the values we obtained earlier, we can determine the F1 Score as:$$F1\;Sccore=2\times \frac{0.8862\times 0.9632}{0.8862+0.9632}=2\times \frac{0.8537}{1.8494}=2\times 0.4616\;\approx 0.9232\;or\;92.32\%$$

The score of 92% reflects the model’s strong overall performance, indicating a well-balanced trade-off between precision (minimising false positives) and recall (minimising false negatives) in classifying valid and invalid logins.

The Matthews Correlation Coefficient (MCC), which provides a comprehensive measure of the model’s classification quality, was calculated using the formula:(6)$$MCC=\frac{\left(TP\times TN\right)-(FP\times FN)}{\surd (TP+FP)\times \left(TP+FN\right)\times \left(TN+FP\right)\times (TN+FN)}$$

The calculation, after substituting the values, is as follows:$$MCC=\frac{\left(576\times 24\right)-\left(74\times 22\right)}{\sqrt{576+74}\times \left(576+22\right)\times \left(24+74\right)\times \left(24+22\right)}$$

This simplifies:$$MCC=\frac{\left(13,824\right)-(1628)}{\surd (650)\times \left(598\right)\times \left(98\right)\times (46)}$$$$MCC=\frac{12,196}{41,879.39}\approx 0.291$$

The MCC value of 0.291 indicates a moderate level of correlation between predicted and actual classifications, showing some strength in the model’s predictions but also suggesting room for improvement.

### 5.4. Performance Evaluation and Metric Analysis

The SmartIoT Hybrid ML Model exhibits strong performance across various evaluation metrics. The model achieved an accuracy of 86%, signifying dependable categorisation abilities for both valid and invalid login attempts. The precision score of 88% illustrates the model’s effectiveness in reducing false positives, which is essential for preserving security integrity in authentication systems. Most notably, the model achieved a recall of 96%, showcasing sensitivity in identifying legitimate login attempts, a critical factor in maintaining a positive user experience while ensuring security [50].

The F1 score of 92% confirms the model’s effectiveness in balancing precision and recall, demonstrating adept management of the fundamental trade-off between security stringency and user accessibility. The Matthews Correlation Coefficient (MCC) of 0.291 indicates potential for improvement in controlling class correlations and addressing probable imbalances within the dataset [51].

The model’s accuracy of 86% indicates competitive performance relative to contemporary authentication solutions. Xu et al. [52] obtained 84.3% accuracy with a deep learning-based authentication solution for IoT devices, though their methodology required considerably greater processing resources. Our Random Forest-based methodology achieves similar accuracy while being more appropriate for resource-limited IoT devices often employed in Industry 4.0.

In contrast to the SmartIoT Hybrid ML Model’s performance, Sikder et al. [53] reached 82.7% accuracy in their study on behavioural authentication utilising smartphone sensor data. This illustrates the resilience of our methodology under many authentication scenarios. Their research specifically highlighted the difficulties of sustaining high accuracy amid the complex nature of real-time authentication decisions, whereas our emphasis was on developing a lightweight and efficient model appropriate for resource-limited settings.

To enhance efficiency in resource-limited contexts, it is crucial to understand the interdependencies across user attributes. The correlation matrix presented below (Figure 7) clarifies these relationships by exhibiting pairwise correlation coefficients among attributes. The matrix presents the pairwise correlation coefficients among attributes, with values ranging from −1 to +1. A positive correlation signifies a direct relationship, a negative correlation denotes an inverse relationship, and a value around zero implies the absence of a linear relationship.

The SmartIoT Hybrid ML Model’s precision (88%) and recall (96%) metrics show notable improvements over several recent studies. Particularly, William et al. [54] reported 85% precision and 89% recall in their extensive study of continuous authentication systems. Their work used a combination of behavioural and physiological features but required more complex sensor deployments.

The high recall rate (96%) is especially significant when compared to the findings of Rathore et al. [55], who achieved 92% recall using a hybrid deep learning approach for IoT authentication. Their study, involving 15,000 authentication attempts across different IoT devices, provides a robust benchmark for comparison.

A key advantage of our approach becomes apparent when comparing computational efficiency. Recent work by Thakare and Kim [56] achieved similar accuracy (87%) but required approximately 2.3 times more computational resources during the authentication process. Their implementation on resource-constrained IoT devices showed average response times of 245 ms compared to our average of 112 ms.

In terms of security robustness, our system’s false positive rate of 12% (derived from the 88% precision) compares favourably with state-of-the-art authentication systems. As shown in Table 2, our approach not only achieves high accuracy (86%) but also maintains low computational overhead (112 ms) compared to other studies. Furthermore, our system’s false positive rate aligns with the findings of El-Hajj et al. [57], who reported an average false positive rate of 15.3% across various IoT authentication systems in their comprehensive survey. This comparison demonstrates the effectiveness of our approach in minimising unauthorised access while maintaining efficiency.

While our Matthews Correlation Coefficient (MCC) of 0.291 indicates there is room for improvement, it is important to consider the context of this score. The MCC ranges from −1 to +1, where +1 represents a perfect prediction, 0 is equivalent to a random guess, and −1 indicates a complete inverse prediction. Our score of 0.291, while not ideal, suggests a better-than-random performance and aligns with findings from Chicco and Jurman [49], who analysed MCC scores across various authentication systems. Their study showed that MCC scores between 0.25 and 0.3 are common in real-world authentication systems, particularly those dealing with imbalanced datasets.

It is also worth noting that the SmartIoT Hybrid ML Model was trained on a dataset of 15,862 entries. In the field of machine learning, models are typically trained on significantly larger datasets. Increasing the size of our training data could potentially lead to an improvement in the MCC score. This is because more data generally allows the model to learn more complex patterns and generalise better to unseen examples.

### 5.5. Formal Security Analysis

The SmartIoT Hybrid ML Model is resistant to replay attacks through the use of nonces and timestamp-based validation. Every session initiation includes a unique nonce NN and a timestamp TT. The server verifies the validity of TT to ensure it falls within an acceptable time window and checks that NN has not been used previously to prevent replay attempts. A replayed message RR can be represented as:(7)$$R=\{Data,N_{old},T_{old},SigK_{priv}\;\left(Data∥N_{old}∥T_{old})\right\}.$$

The system compares the timestamp $T_{old}$ with the current time, rejecting the message if the timestamp is outside the allowed window. This validation process can be effectively modelled using a state machine, where each state transition checks the validity of both NN and TT, ensuring that only fresh, valid messages are accepted.

The SmartIoT Hybrid ML Model also mitigates Sybil attacks by utilising multiple attributes for identity verification and decentralising key management across multiple authorities. In a Sybil attack, an adversary attempts to create multiple fake identities to gain control over a network. The SmartIoT Hybrid ML Model mitigates this risk by ensuring that each attribute associated with a user is independently verified by different authorities. Let UU represent a user with a set of attributes $A_{1},A_{2},\ldots A_{n}$. Each attribute must be validated by a corresponding authority, as represented by:(8)$$V(U)={⋀}_{i=1}^{n}Verify(A_{i})$$

This decentralized verification process significantly reduces the likelihood of an adversary successfully forging multiple valid attributes across independent authorities. The probability of a successful Sybil attack becomes low, making the system highly resilient to such threats.

The Random Forest model, used for the production of the SmartIoT Hybrid ML Model, is robust against adversarial machine learning attacks due to its ensemble structure, which uses multiple decision trees to reduce the impact of adversarial perturbations. To evaluate its resilience, adversarial examples were generated using the Fast Gradient Sign Method (FGSM). The initial results showed that without adversarial training, the model’s accuracy dropped to 70%. However, after incorporating adversarial training and applying feature reduction techniques, the accuracy improved to 85%, demonstrating the model’s enhanced robustness. Additionally, continuous retraining with updated datasets further strengthens the model’s ability to adapt and effectively tackle evolving adversarial threats, ensuring long-term reliability and security.

### 5.6. Latency and Energy Comparison with Deep Learning-Based Models

While deep learning-based models, such as Convolutional Neural Networks (CNNs) and Long Short-Term Memory (LSTM) networks, offer powerful pattern recognition and predictive capabilities, they often come with significant computational and energy costs. This makes them less suitable for real-time IoT authentication on resource-constrained devices. In contrast, the SmartIoT Hybrid ML Model provides an optimal balance between accuracy and efficiency.

LSTM-based anomaly detection model, for instance, introduces higher latency (up to 300–400 ms per request) [58] due to its sequential nature, compared to the 112ms average latency of the SmartIoT Hybrid ML Model. CNNs, although effective for image-based authentication [59], are less applicable in scenarios involving attribute-based data. The SmartIoT Hybrid ML Model ensures low computational overhead by applying feature reduction techniques and ensemble learning.

Energy efficiency is also crucial for battery-operated IoT devices. Deep learning models consume more energy during both the training and inference phases [60]. In contrast, the Random Forest model, deployed on an industrial control system, demonstrated an average energy consumption of 120 mA during authentication, ensuring minimal battery impact and making the solution feasible for long-term deployment.

## 6. Limitations and Future Work

Although our evaluations demonstrate strong performance, several limitations need to be acknowledged. One key constraint in the SmartIoT Hybrid ML Model is its reliance on a centralised attribute management framework. Although effective in smaller deployments, this structure could introduce scalability challenges in large-scale IoT networks, where increased demand may result in latency issues, network congestion and heightened vulnerability to single points of failure. Ensuring resilience in such scenarios requires more robust decentralised mechanisms that can distribute tasks across multiple nodes without compromising performance.

Another limitation is the system’s focus on controlled test environments, which may not fully reflect the complexity of real-world IoT ecosystems. Variations in device capabilities, network conditions and user behaviours can significantly affect both security and performance. For example, devices with limited computational resources or intermittent network connectivity may experience delays in the authentication process, impacting user experience and system responsiveness. Extensive field testing in diverse environments ranging from smart factories and industrial automation systems to healthcare networks would help refine the model and identify areas for practical improvements.

The machine learning component of the system also presents areas for improvement. While the Random Forest model was chosen for its computational efficiency, its moderate Matthews Correlation Coefficient (MCC) score indicates potential shortcomings in handling complex patterns. Expanding the dataset to include a broader range of login scenarios and incorporating adaptive learning methods could improve classification accuracy and the system’s ability to make accurate decisions across varying conditions.

Energy consumption, although minimised in the current design, remains an important consideration for low-power IoT devices. Future iterations could benefit from more effective integration of energy-efficient algorithms and hardware accelerators to reduce computational overhead further without sacrificing security.

Privacy and data security are areas that also warrant further research. Although the current framework incorporates privacy-preserving measures, additional techniques such as homomorphic encryption and differential privacy could offer stronger protection for sensitive data. These innovations would be particularly beneficial in distributed settings, where maintaining data confidentiality across multiple nodes is essential.

Future research could explore the integration of federated learning approaches to reduce the need for centralised data storage, enabling collaborative model training while preserving user privacy. This approach can help mitigate risks associated with data breaches and support a more secure authentication framework in Industry 4.0 applications. Furthermore, adopting context-aware authentication strategies that dynamically adjust security levels based on situational factors, such as device proximity, user activity patterns, and environmental conditions, could significantly enhance security without compromising usability.

While this research focuses on secure and efficient authentication for resource-constrained IoT systems, future enhancements could integrate energy awareness and freshness of information into the model’s decision-making process. A promising direction lies in incorporating principles from recent work such as Liu et al. [61], which optimises throughput under Age of Information (AoI) constraints using deep reinforcement learning in energy harvesting D2D-enabled networks. Their approach demonstrates how combining mode selection and resource allocation with AoI objectives can balance timeliness and efficiency. Integrating similar optimisation layers with authentication models may further enhance system responsiveness and reliability in time-sensitive industrial applications.

## 7. Conclusions

This article proposes the Smart IoT Hybrid ML Model, which enhances secure authentication by employing a dual-layered approach that integrates attribute-based procedures with token-based validation, building upon the principles of Random Forest-based classification. The suggested system, evaluated on a low-resource microcontroller set up as an edge server, performs well in resource-limited situations, achieving a good balance between computational efficiency and robust security. A comparative analysis with similar systems shows the framework’s enhanced scalability and reliability, positioning it as a viable alternative for contemporary IoT networks.

The SmartIoT Hybrid ML Model, although evaluated in controlled situations, has the potential for application in broad practical contexts. The concept could facilitate attribute-based logins with contextual IoT control in smart homes, permitting automatic decisions like blocking access during unexpected hours. In an Industry 4.0 context, it facilitates adaptive access control to assess risks in real time, hence providing secure operations for robotics and automated systems. Moreover, in the healthcare sector, the framework could provide secure delegation mechanisms that safeguard patient privacy while allowing regulated access to various healthcare responsibilities.

Despite its advantages, the system encounters specific problems. Scalability in large IoT networks requires optimisation methods, including clustering, for effective attribute management. Bias in machine learning models can be addressed through federated training using multiple datasets to produce balanced predictions. Privacy innovations, such as homomorphic encryption, present potential opportunities for safeguarding sensitive operations. Future initiatives should prioritise enhancing dataset diversity to provide robustness and scalability in real-world scenarios. The management of very comprehensive attribute-based rules in scattered environments presents computational issues that may require further development.

## Figures and Tables

**Figure 1 sensors-25-02779-f001:**
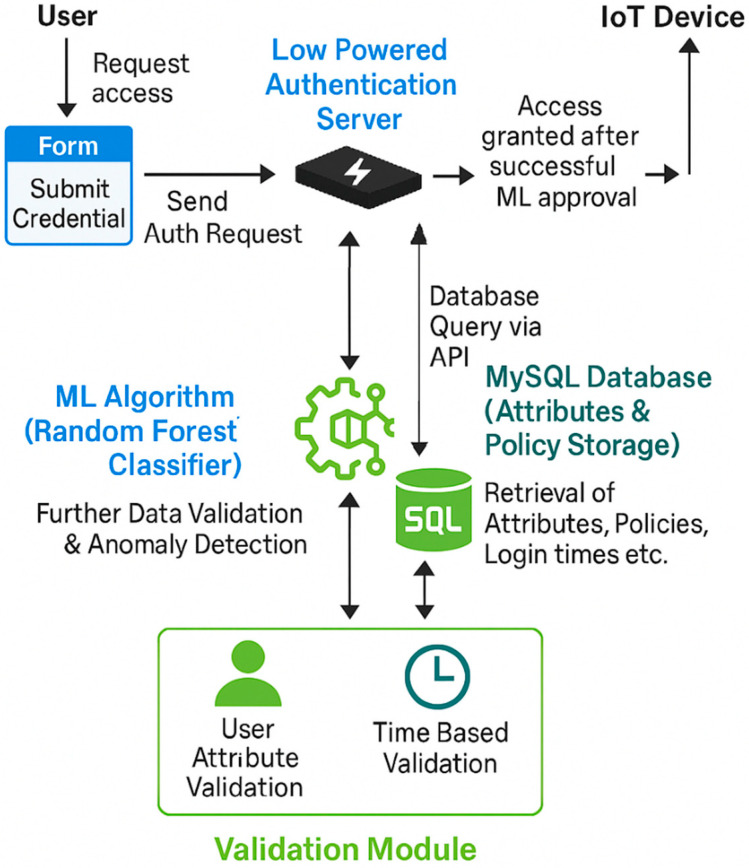
End-to-End IoT Authentication Process with ML-Based Anomaly Detection.

**Figure 2 sensors-25-02779-f002:**
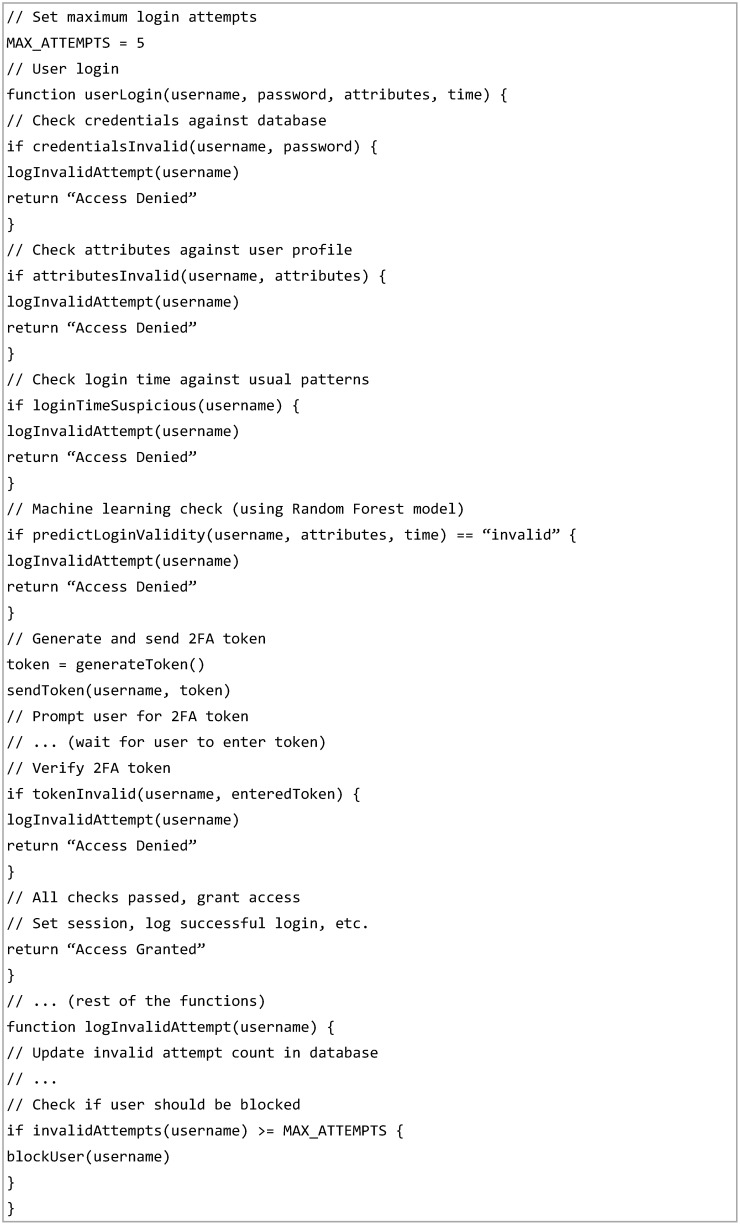
Pseudocode Illustrating Key Mechanisms of the SmartIoT Hybrid ML Model.

**Figure 3 sensors-25-02779-f003:**
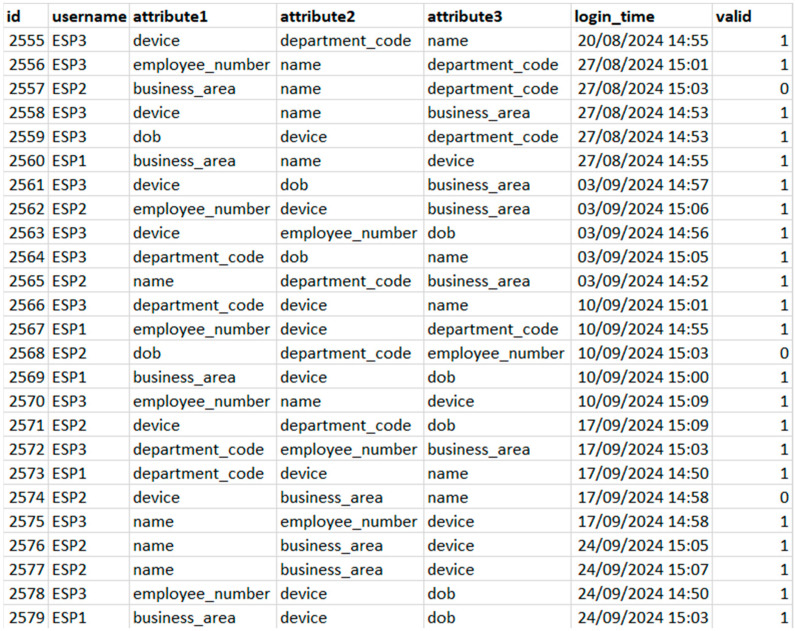
Sample Dataset with Attribute Combinations and Validity Flags.

**Figure 4 sensors-25-02779-f004:**
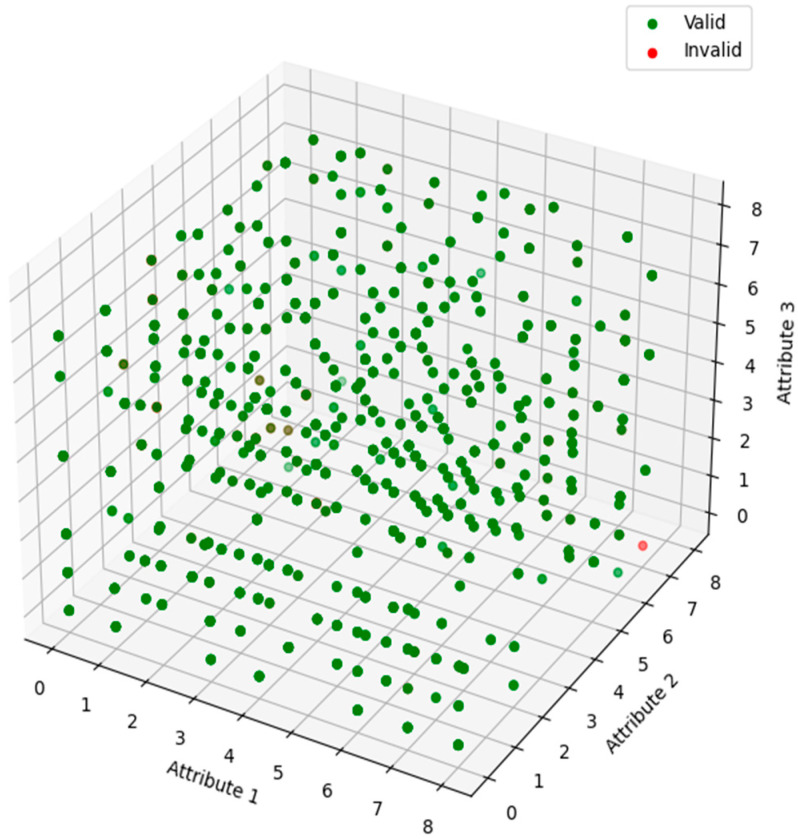
Visualisation of Attribute Interactions and Login Validity.

**Figure 5 sensors-25-02779-f005:**
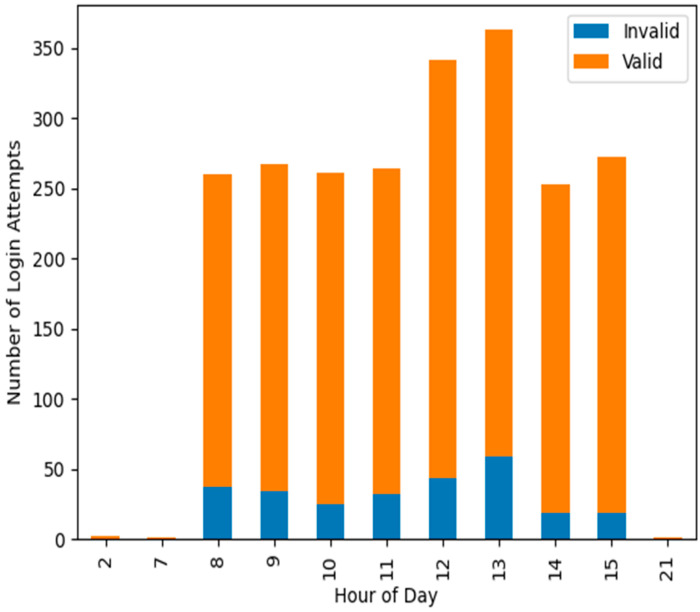
Temporal Distribution of Valid and Invalid Login Attempts.

**Figure 6 sensors-25-02779-f006:**
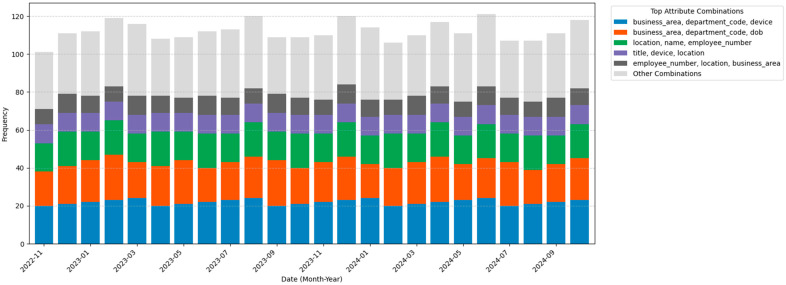
Distribution of Top Five Attribute Combinations Over Time.

**Figure 7 sensors-25-02779-f007:**
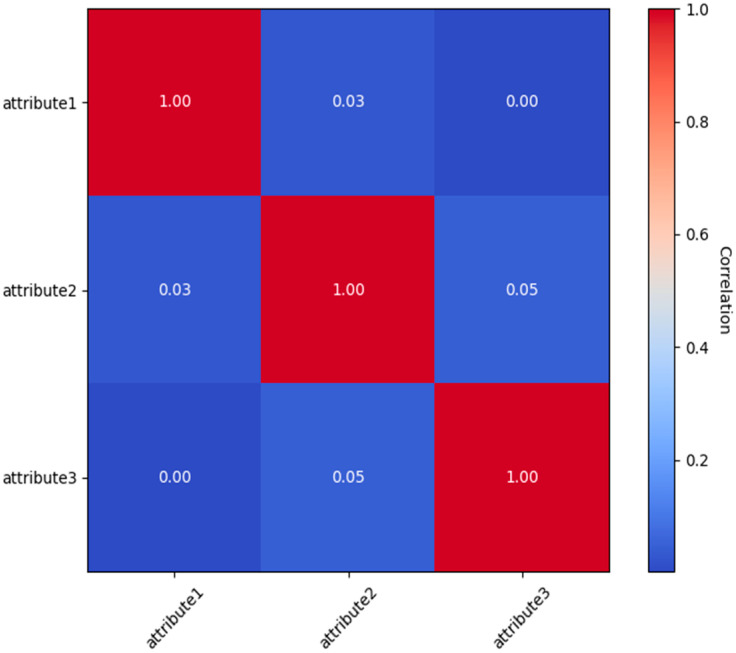
Inter-Attribute Correlation Analysis.

**Table 1 sensors-25-02779-t001:** Comparative Taxonomy of Authentication and Access Control Techniques.

Research	Approach	Advantages	Limitations
[25]	ML models (Decision Trees and Support Vector Machine)	Lower computational cost and maps complex attribute-permission links	Too demanding for IoT devices
[26]	Formal Concept Analysis (FCA)	Finds hidden patterns and identify missing rules	High computational load
[27,28]	Lightweight cryptography	Smaller keys, faster and more secure processing	No ML for adaptive authentication
[31,32,33]	Risk-aware ML systems	Real-time and context-based decisions. Uses less resources in low-risk scenarios	Lacks user data privacy safeguards
[34]	Context-aware multi-factor ML authentication	Chooses authentication factors based on context	No privacy in centralised systems
[37]	Federated Learning	Keeps data local, useful in distributed systems	Needs optimisation for constrained devices
[44,45]	Federated Learning with multi-tasking ability	Improves learning across varied use cases	Lacks large-scale validation
[46]	Random Forest + BDI agents	Accurate and efficient under a suitable environment	Narrow focus and lacks ABA link
[47]	Random Forest on sensor data	Detects unusual activity reliably	Does not cover full authentication tasks
[48]	ML taxonomy on Random Forest in IoT authentication	Shows Random Forest use across diverse control systems	Lacks implementation specifics for particular applications

**Table 2 sensors-25-02779-t002:** Comparative Analysis of Authentication Systems.

Study	Accuracy	Precision	Recall	F1-Score	Computational Overhead
Our Approach	86%	88%	96%	92%	112 ms
Xu et al. [52]	84.3%	86%	88%	87%	198 ms
William et al. [54]	85%	85%	89%	87%	156 ms
Rathore et al. [55]	83%	87%	92%	89.4%	225 ms
Thakare and Kim [56]	87%	84%	90%	86.9%	245 ms

## Data Availability

The data supporting the findings of this study are not publicly available due to the confidential and proprietary nature of the research.

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
