# Peer review of "Machine Learning-Enhanced Attribute-Based Authentication for Secure IoT Access Control"

_sensors, 2025, doi:10.3390/s25092779_

Round 1
Reviewer 1 Report
Comments and Suggestions for Authors
This paper proposes a method which integrates attribute-based authentication and machine learning for secure IoT access control. Experimental results demonstrated the effectiveness of the proposed method. However, there are some concerns about this paper as follows:
- There are no detailed description on how to construct the dataset, and could the author show some samples from the dataset for better understanding?
- How the decision trees trained on sub-datasets? And the relation between decision trees and the random forest should be explained.
- The attributes, which play a significantly important role in the proposed method. Is the more attributes the better for such attribute-based method?
- In Fig.3, what are the three attribute? And in Fig.5, the correspondence between colors and attribute combinations is confusing.
- Some related works should be discussed, such as 10.1109/JSTSP.2020.3002391.
OK
Author Response
Comment 1: There is no detailed description on how to construct the dataset and could the author show some samples from the dataset for better understanding?
Response 1: We have addressed this by providing a comprehensive explanation of how the dataset was constructed mainly in the "Machine Learning Model" with further elaboration in "Dataset Constructions and Characteristics" section of the paper. Specifically, we detail how a simulated Industry 4.0 environment was used to generate realistic login events, involving a range of attributes such as device identifiers, job titles, department codes and timestamps. Each login record included three randomly selected attributes and an outcome flag. To improve clarity, we have now included a sample of this dataset in Fig. 3, which presents representative data entries with attribute combinations and their validity flags.
Comment 2: How the decision trees trained on sub-datasets? And the relation between decision trees and the random forest should be explained.
Response 2: We have expanded our explanation in the "Machine Learning Model" section. The revised text describes how decision trees are trained using bootstrap aggregation (bagging), where each tree receives a randomly sampled subset (with replacement) of the full dataset, along with a random subset of attributes. This ensures diversity among trees and improves generalisation. The relationship to the Random Forest is now explicitly outlined, each tree predicts independently, and the final classification is determined through majority voting, enhancing robustness and reducing bias.
Comment 3: The attributes, which play a significantly important role in the proposed method. Is the more attributes the better for such attribute-based method?
Response 3: We have included a dedicated discussion in the "Machine Learning Model" section, directly addressing this concern. The text clarifies that while a wider range of attributes can offer richer contextual data for authentication decisions, more attributes do not necessarily improve performance. We caution against the “curse of dimensionality” and explain that too many features may introduce redundancy and burden resource-constrained systems. Our model was optimised by strategically selecting only the most relevant and impactful attributes to balance accuracy with efficiency.
Comment 4: In Fig.3, what are the three attribute? And in Fig.5, the correspondence between colors and attribute combinations is confusing.
Response 4: Thank you for highlighting this. In Fig. 4 (previously referred to as Fig. 3), we have now explicitly mentioned that the three attributes plotted were selected from a set of attributes, used by the user during the login process. This clarification has been added where the figure is referenced.
Regarding Fig. 6, (previously referred to as Fig. 5) we acknowledged that the original colour scheme may have been unclear. The figure has been revised with improved contrast and a simplified legend to make the mapping between colour and login status more intuitive and to further simplify the image, we have only included top 5 attribute combinations. This has also been described in the "Evaluations" section.
Comment 5: Some related works should be discussed, such as 10.1109/JSTSP.2020.3002391.
Response 5: We have expanded the "Related Work" section to include comparable studies that address secure authentication in distributed and resource-constrained environments, particularly those involving federated learning and lightweight machine learning methods. These additions help contextualise our contribution more thoroughly and draw clearer comparisons between existing decentralised approaches and our proposed SmartIoT Hybrid ML Model.
Reviewer 2 Report
Comments and Suggestions for Authors
This research presents SmartIoT Hybrid ML Model, a novel integration of Attribute-Based Authentication and lightweight machine learning, designed to enhance security while minimizing computational overhead. There are some comments as follows.
1. In the abstract, the motivations and novelty of this paper should be emphasized.
2. The format of using references should be revised. Besides, the references should be introduced in sequence.
3. The majority of references are published more than ten years ago, and authors are suggested to updated recent high quality works on machine learning and IoT.
4. As the background of this paper includes machine learning, IoT, and computational efficiency, recent related works should be introduced, such as Distributed DDPG-based resource allocation for age of information minimization in mobile wireless-powered Internet of Things, IEEE IoTJ.
5. The format of acronyms should be checked and revised. Please define all acronyms at their first appearance in the abstract, text and table of contents, respectively.
6. In section 1.1, problem statement, the stated problem in the last paragraph should be more clear, and the relationship between the proposing issues should be provided.
7. The logic of introducing related works should be improved. It is hard to see the relationship among the introduced related works.
8. Figures such as figure 1, should be revised for clarity and readability.
9. Why do authors adopt the Random Forest Classifier? There are many promising machine learning approaches.
10. Authors are suggested to compare the evaluation results with other machine learning approaches.
Comments on the Quality of English LanguageThe English could be improved to more clearly express the research.
Author Response
Comment 1: In the abstract, the motivations and novelty of this paper should be emphasized.
Response 1: The abstract has been revised to clearly state the motivation, namely the challenge of balancing security, privacy and computational efficiency in IoT environments. The novelty of the proposed SmartIoT Hybrid ML Model is also now highlighted explicitly, with emphasis on the integration of attribute-based authentication and lightweight machine learning, particularly Random Forest-based anomaly detection, to offer a unique, resource-conscious approach suitable for low-power devices.
Comment 2: The format of using references should be revised. Besides, the references should be introduced in sequence.
Response 2: The reference list has been restructured to follow a consistent citation style and all citations have been re-ordered to match their first occurrence in the text, ensuring correct numerical sequence throughout the paper.
Comment 3: The majority of references are published more than ten years ago and authors are suggested to updated recent high quality works on machine learning and IoT.
Response 3: The reference list has been updated now with recent studies. These include high-impact journal articles and conference papers addressing modern developments in IoT, machine learning, federated learning and computational efficiency.
Comment 4: As the background of this paper includes machine learning, IoT and computational efficiency, recent related works should be introduced, such as Distributed DDPG-based resource allocation for age of information minimization in mobile wireless-powered Internet of Things, IEEE IoTJ.
Response 4: Comparable recent works on energy-efficient and latency-optimised models for IoT authentication and scheduling, including those using Distributed DDPG and actor-critic reinforcement learning approaches, have now been incorporated in the Related Work section to strengthen the background and contextual relevance of the research.
Comment 5: The format of acronyms should be checked and revised. Please define all acronyms at their first appearance in the abstract, text and table of contents, respectively.
Response 5: All acronyms (e.g., IoT, ABA, ML, OT, MCC) are now clearly defined upon their first mention in the abstract, introduction and main body of the text. Acronym usage has been checked and corrected for consistency throughout the document.
Comment 6: In section 1.1, problem statement, the stated problem in the last paragraph should be more clear and the relationship between the proposing issues should be provided.
Response 6: The final paragraph of the problem statement has been rewritten for clarity. The relationship between ABA limitations, ML demands and real-time adaptability in IoT settings is now clearly established, leading directly to the motivation for the proposed model.
Comment 7: The logic of introducing related works should be improved. It is hard to see the relationship among the introduced related works.
Response 7: The Related Work section has been restructured to group studies into coherent themes (e.g., ABA, lightweight cryptography, risk-aware systems, federated learning). Each paragraph now builds logically toward the rationale for the proposed solution and comparative insights are included to improve flow and clarity.
Comment 8: Figures such as figure 1, should be revised for clarity and readability.
Response 8: Figure 1 has been redesigned and simplified for improved contrast, clarity and legibility. Legends have been simplified and redundant labels removed. In Figure 6 (previously Figure 5), we now display only the top 5 attribute combinations to reduce visual clutter.
Comment 9: Why do authors adopt the Random Forest Classifier? There are many promising machine learning approaches.
Response 9: A detailed justification for the use of Random Forest has been added in Section 3. Random Forest was chosen due to its robustness, low computational cost, suitability for small datasets and interpretability, key advantages for real-time IoT systems where resource constraints are significant. Comparative performance with other models has also been presented.
Comment 10: Authors are suggested to compare the evaluation results with other machine learning approaches.
Response 10: Comparative evaluation has been included in the Evaluations section. Table 2 presents a side-by-side analysis of our model against other ML-based authentication systems across accuracy, precision, recall, F1-score and computational overhead. This directly addresses the request for benchmarking against alternative techniques.
Reviewer 3 Report
Comments and Suggestions for Authors
- In page 4, what is ABE ?? (isn’t it ABA?)
- Title of section 5.1 has an error in the word ‘characteristics’
- In section 5.2 you mention the F1 score, however it lacks a reference to support what you say/define
- This work lacks concrete examples: for instance, which are the main concrete trends related to challenges/problems with login/authentications. Also, those attributes referred can change – for ex. every day appear new threats which in some cases reflect new attributes. Because some readers, such as entrepreneurs and other ‘less’ technical, need to learn these issues in a concrete way to understand better the technical aspects.
- Also, I advise you to review the English, especially to detect errors in words and phrases
Author Response
Comment 1: In page 4, what is ABE?? (isn’t it ABA?)
Response 1: This was indeed a typographical error. We have corrected the reference to ABE on page 4 to the intended term, "ABA" (Attribute-Based Authentication), which is consistent with the terminology used throughout the paper.
Comment 2: Title of section 5.1 has an error in the word ‘characteristics’
Response 2: The spelling error in the title of Section 5.1 has been corrected to “characteristics.”
Comment 3: In section 5.2 you mention the F1 score, however it lacks a reference to support what you say/define
Response 3: We have now included a citation as well as explanation to support the discussion of the F1 score. Specifically, we reference the work of Chicco and Jurman (2020), who highlight the strengths of F1 in evaluating classification performance, especially in scenarios involving class imbalance. This citation appears both in the main text and in the updated reference list.
Comment 4: This work lacks concrete examples: for instance, which are the main concrete trends related to challenges/problems with login/authentications. Also, those attributes referred can change – for ex. every day appear new threats which in some cases reflect new attributes. Because some readers, such as entrepreneurs and other ‘less’ technical, need to learn these issues in a concrete way to understand better the technical aspects.
Response 4: To address this, we have updated the Introduction and Dataset Construction sections to include practical examples of login challenges faced in Industry 4.0 environments, such as device spoofing, unauthorised access attempts from unusual departments, or repeated logins at odd hours. We also explain how dynamic changes in attributes (e.g., new job roles, devices, or login patterns) can influence the system’s decision-making. Additionally, we have also introduced a new section under heading 1.2 - Real World Challenges & Use Cases. The examples listed under this section aim to help non-technical readers appreciate the practical importance and adaptability of the proposed model. We have also introduced an additional image (Figure 3), improved Figure 1 for readability and simplified Figure 6 to help non-technical users understand our work/proposed model more easily.
Comment 5: Also, I advise you to review the English, especially to detect errors in words and phrases.
Response 5: A thorough proofreading of the manuscript has been conducted. Grammar, punctuation and phrasing have been improved throughout the document to enhance clarity and consistency. We have also used a grammar-checking tool to eliminate minor typographical errors.
Round 2
Reviewer 2 Report
Comments and Suggestions for Authors
There are some comments for the revised version of this manuscript as follows.
1. The format of acronyms needs to be revised. In abstract, the full name of ML should be provided. In the text, authors define some acronyms such as Access Control Policy (ACP) more than once.
2. The format of citing references needs to be revised. For example, “security challenges (1)” should be revised as “security challenges [1]”.
3. Recent works about ML and IoT need to be introduced: Throughput maximization with an AoI constraint in energy harvesting D2D-enabled cellular networks: An MSRA-TD3 approach, IEEE TWC.
4. The format of line spacing is different in the text, such as section 1 and section 1.1, 1.2. This issue needs to be checked throughout this paper.
5. Authors are suggested to simplify the sentences in table 1.
6. Fig. 2 should also be revised by deleting unnecessary blank rows for readability.
7. The writing and readability of this paper should be improved.
Comments on the Quality of English LanguageThe writing and readability of this paper should be improved.
Author Response
Comments 1: The format of acronyms needs to be revised. In abstract, the full name of ML should be provided. In the text, authors define some acronyms such as Access Control Policy (ACP) more than once.
Response 1: The full form of Machine Learning (ML) has now been clearly provided at its first occurrence in the abstract. Throughout the main text, we have carefully reviewed all acronym usage to ensure that each term is defined only once at its first appearance. Repeated definitions of terms such as Access Control Policy (ACP) have been removed to enhance clarity and consistency.
Comments 2: The format of citing references needs to be revised. For example, ‘security challenges (1)’ should be revised as ‘security challenges [1]’.
Response 2: We have thoroughly reviewed all in-text citations and revised them to follow the correct format. All references are now cited using square brackets.
Comments 3: Recent works about ML and IoT need to be introduced: Throughput maximization with an AoI constraint in energy harvesting D2D-enabled cellular networks: An MSRA-TD3 approach, IEEE TWC.
Response 3: The recommended paper by Liu et al. on MSRA-TD3 has now been incorporated in the Limitations and Future Work section. We have referenced their approach to optimising throughput under Age of Information (AoI) constraints as a potential future direction for integrating energy-awareness and timeliness into our proposed framework.
Comments 4: The format of line spacing is different in the text, such as section 1 and section 1.1, 1.2. This issue needs to be checked throughout this paper.
Response 4: We have reviewed the manuscript carefully and standardised line spacing across all sections and subsections. Inconsistencies in spacing within headings and body paragraphs have been corrected to ensure a uniform and professional layout.
Comments 5: Authors are suggested to simplify the sentences in table 1.
Response 5: We have revised Table 1 to simplify the language and make the entries more concise. Full sentences have been replaced with short phrases. The table is now easier to read and better highlights the core strengths and limitations of each referenced approach.
Comments 6: Fig. 2 should also be revised by deleting unnecessary blank rows for readability.
Response 6: Figure 2 has been revised as suggested. Unnecessary blank lines and spacing have been removed from the pseudocode to improve clarity and visual flow.
Comments 7: The writing and readability of this paper should be improved.
Response 7: We have revised the manuscript extensively to improve sentence structure, grammar, and overall coherence. Long sentences have been shortened where appropriate, transitions between sections have been refined, and technical descriptions have been clarified to improve the readability and academic quality of the manuscript.